# Plant leaves inspired sunlight-driven purifier for high-efficiency clean water production

Hongya Geng[1], Qiang Xu[2], Mingmao Wu[1], Hongyun Ma [1], Panpan Zhang[3], Tiantian Gao[1], Liangti Qu[1,2,3], Tianbao Ma[2] & Chun Li [1]

Natural vascular plants leaves rely on differences in osmotic pressure, transpiration and guttation to produce tons of clean water, powered by sunlight. Inspired by this, we report a sunlight-driven purifier for high-efficiency water purification and production. This sunlight-driven purifier is characterized by a negative temperature response poly(N-iso-propylacrylamide) hydrogel (PN) anchored onto a superhydrophilic melamine foam skeleton, and a layer of PNIPAm modified graphene (PG) filter membrane coated outside. Molecular dynamics simulation and experimental results show that the superhydrophilicity of the relatively rigid melamine skeleton significantly accelerates the swelling/deswelling rate of the PNPG-F purifier. Under one sun, this rational engineered structure offers a collection of 4.2 kg m$^{-2}$ h$^{-1}$ and an ionic rejection of > 99% for a single PNPG-F from brine feed via the cooperation of transpiration and guttation. We envision that such a high-efficiency sunlight driven system could have great potential applications in diverse water treatments.

---

[1] MOE Key Laboratory of Bioorganic Phosphorus Chemistry & Chemical Biology, Department of Chemistry, Tsinghua University, Beijing 100084, P. R. China. [2] State Key Laboratory of Tribology, and Key Laboratory for Advanced Materials Processing Technology, Ministry of Education of P. R. China, Department of Mechanical Engineering, Tsinghua University, Beijing 100084, P. R. China. [3] School of Chemistry and Chemical Engineering, Beijing Institute of Technology, Beijing 100081, P. R. China. Correspondence and requests for materials should be addressed to L.Q. (email: lqu@mail.tsinghua.edu.cn) or to T.M. (email: mtb@mail.tsinghua.edu.cn) or to C.L. (email: chunli@mail.tsinghua.edu.cn)

Taking full utilization of sunlight for clean water production would enable instructive directions of research in a wide range of environmental, agricultural, and ecological fields[1–4]. Efforts have been devoted into solar steam generation systems for clean water production[5–7]. However, the theoretical water collection rate is limited to be of a low value and a large amount of energy is necessary for condensing freshwater[8,9]. More seriously, the salt deposition on the photothermal materials cannot be avoided effectively. That is a common occurrence that further lowers the water collection rate in solar steam generation systems. Membrane-based technology, on the other hand, is suitable in case of brine condition[10]. Unfortunately, vast quantities of electric or fuel energy are consumed in practical usage. These two methods have a difficulty to meet the increasing requirement of water treatment with high efficiency and low-energy consumption.

Natural plant leaves transport tons of water from the soil per day depending on a specialized phloem network powered by sunlight. The loading of sugars and ions within the semipermeable plasmodesmata porous interface lowers the chemical potential of water, allowing continuous water suction[11]. Subsequently, the stomata on their leaves transpire over daytime and exude water at night through a special hydathode of a leaf tip or edge that is called guttation[12–15]. The mechanism by which plants leaves transport clean water underground to mature leaves, is the cooperation of transpiration and guttation effect that allows for continuous clean water production. The loss of water by transpiration and guttation reduces pressure in the vascular plant, pulling the continuous transport of clean water with high efficiency[16]. The integration of this unique physiology of natural plant leaves leads to the design and development of classes of water purification devices.

Inspired by natural plant leaves, we develop a highly efficient sunlight-driven clean water purifier to significantly improve the water collection rate from brine feed or organic pollutes. The promoted performance arises from the combinations of the transpiration and guttation. As shown in Fig. 1a, the sunlight-driven bulk water purifier is constructed by growing poly (N-isopropylacrylamide) (PNIPAm) hydrogels (PN) on the surface of a superhydrophilic melamine foam skeleton (PN-F), followed by coating this foam with PNIPAm-modified graphene (PG) nanofiltration membrane (PNPG-F). The PN core exhibits a rapid, self-propelled, and reversible water suction and guttation properties, due to the swelling/deswelling switch near the lower critical solution temperature (LCST)[17]. The superhydrophilic melamine foam is employed to accelerate the water suction and guttation rate of PN[18,19]. The PG membrane endows it the capability to separate ions and molecules with a relatively high rejection rate (> 99%), which allows the purifier to absorb clean water in brine feed or organic pollutes. Due to the excellent sunlight-to-heat transformation of PG and the unique thermal-responsive feature of PN, clean water is produced via transpiration and guttation (Fig. 1b). Consequently, the water collection rate reaches $4.2\,kg\,m^{-2}\,h^{-1}$ under 1-sun irradiation, which represents the highest yields powered by sunlight. This sunlight-powered self-propelled purifier retains good responsiveness and high-efficiency water purification ability for a long-term use.

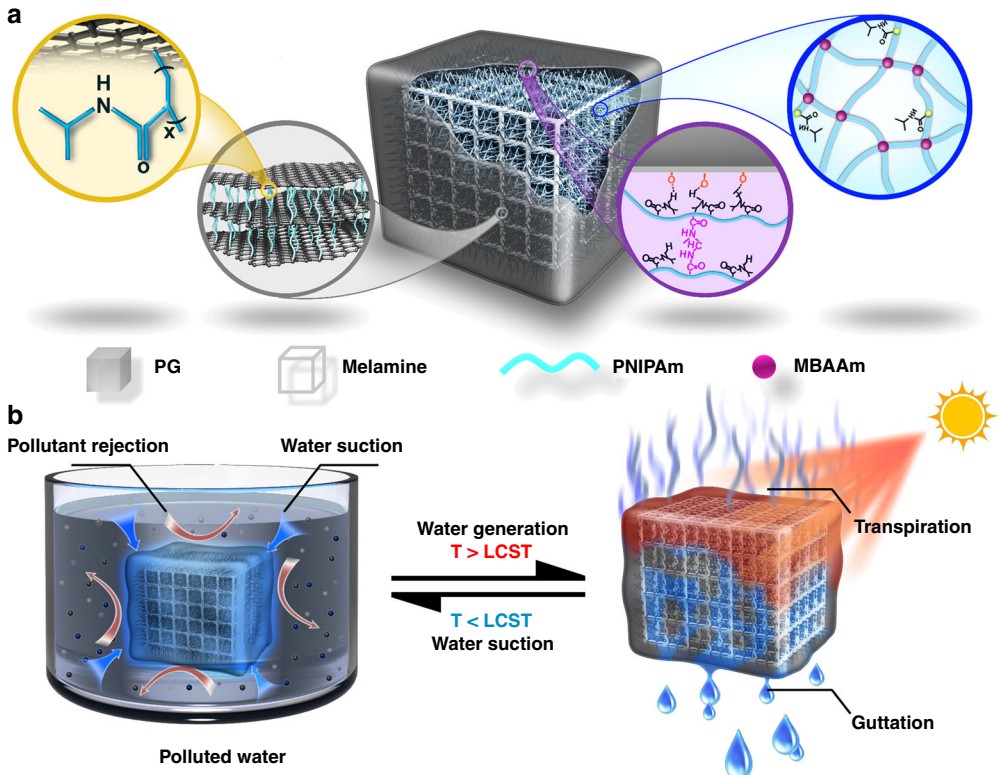

**Fig. 1** Schematic of water treatment of the sunlight-powered purifier (PNPG-F). **a** Microstructure of the sunlight-driven purifier. The PNPG-F purifier consists of PNIPAm chains growing on the surface of melamine skeleton and PG membrane coated outside. Melamine provides a water suction channel via capillary force and inhibits bulk volume collapse of the purifier. **b** Water purification procedure based on the PNPG-F purifier. The coated PG demonstrates solar–thermal conversion and pollute rejection. After being immersed in pollute water, PNPG-F adsorbs a large amount of clean water. Under sunlight irradiation, the PG membrane converts sunlight into thermal energy and heats PN to the temperature above LCST, facilitating hydrophilization; thus, clean water can be produced via transpiration and guttation

## Results

**Fabrication of the sunlight-driven PNPG-F purifier**. The PNPG-F purifier consists of a commercially available melamine foam, PNIPAm hydrogel (PN), and N-isopropylacrylamide (NIPAm) modified graphene (PG). Briefly, for preparing the sunlight-driven PNPG-F purifier, an initiator and accelerator (ammonium persulfate, APS and N,N,N,N′-tetramethyl-ethyl-enedianine, TEMED) were first sprayed homogeneously onto a melamine skeleton by an atomizer (Fig. 2a). Controllable polymerization and gelation of N-isopropylacrylamide (NIPAm) and N,N′-methylenebis(acrylamide) (MBAAm) is initiated by APS on the surface of the melamine skeleton (Supplementary Figure 1). Simultaneously, PG was prepared by free-radical polymerization via mixing NIPAm, 2-2′-azobis(2-methylpropionitrile), and graphene oxide in dimethylformamide. Finally, a solution containing 5.0 mg mL$^{-1}$ of PG was casted outside the foam, followed by annealing at 50 °C overnight (Supplementary Figure 2 and 3). The whole procedure was repeated, except for the employment of the melamine skeleton to prepare PNPG.

The freeze-dried PNPG-F purifier displayed a honeycomb-like structure with nearly dense cell walls as reported previously (Fig. 2b, c and Supplementary Figure 3)[20]. Melamine foam used in this work is a commercially available product rich in amino groups. As shown in N1s spectrum of melamine foam and PN (Supplementary Figure 4 and 5), two nitrogen-binding states in the case of melamine foam were identified with binding energies of 400.5 (pyrrole or pyridone) and 398.4 eV (C–N=C), respectively[21]. On the other hand, the N1s spectrum of PNIPAm exhibits the existence of N−C=O (≈399.4 eV), forming the strong hydrogen bonds between the amino group of melamine and acylamino of PNIPAm, as well as Van der Waals' force.

For the outside coating layer, a highly stacked PG membrane could be found. The successful modification of graphene oxide (GO) nanosheets by PNIPAm was demonstrated by the results of Fourier-infrared spectra. Several new peaks at 1642, 1540, and 1367–1460 cm$^{-1}$ appeared after grafting with PNIPAm (Supplementary Figure 6a), which are consistent with the characterizations reported previously[22]. Furthermore, the intensity ratio between D and G ($I_D/I_G$) increases from 0.91 to 1.03. This change demonstrated that the modification of PNIPAm on the surface of GO slightly breaks C=C bonds (Supplementary Figure 6b and 7). The 1H NMR spectroscopic study of the PG demonstrated the successful modification (Supplementary Figure 8). After the PNIPAm was grafted on the surface of GO nanosheets, the proton peak of CH$_3$ (a), CH (b), CH$_2$ (c), and CH (d) groups appeared (Supplementary Figure 9)[23]. We further characterized the thickness of PG and GO nanosheets. The higher height of PG (~4.2 nm) than that of GO nanosheets clearly suggested the successful modification of PNIPAm chains (Supplementary Figure 10).

The addition of PG and melamine skeleton altered energy storage and energy dissipation of PN, which can be represented

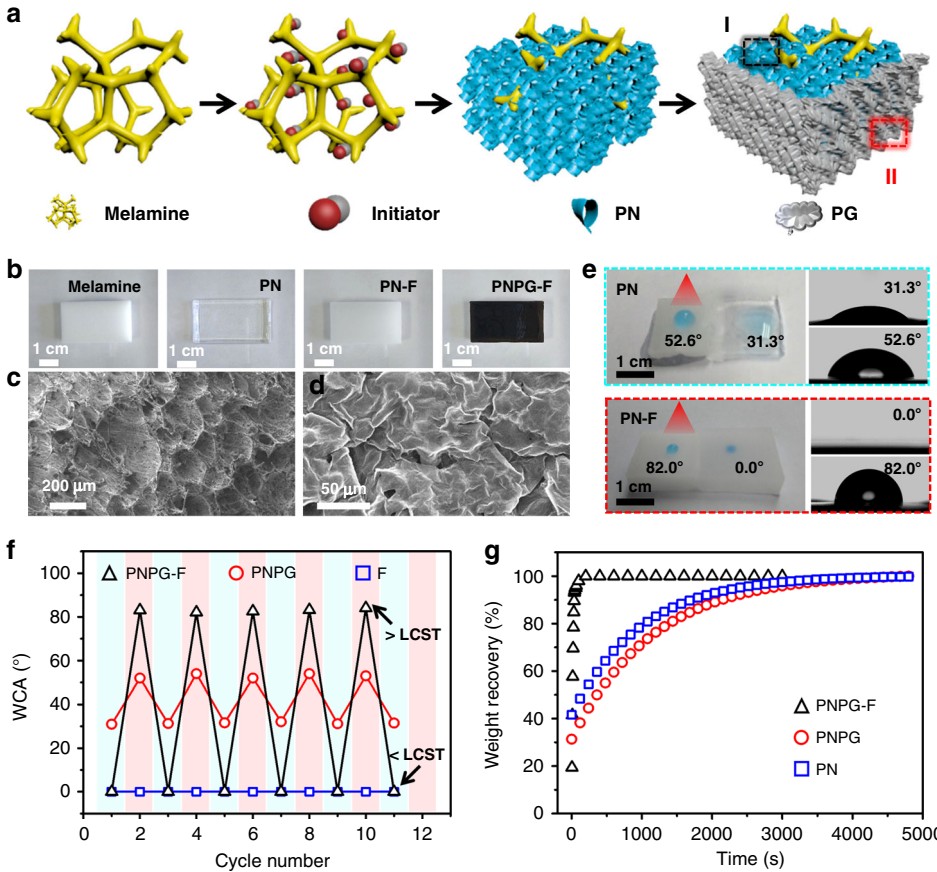

**Fig. 2** Rapid weight recovery of a PNPG-F purifier. **a** Preparation procedure of a PNPG-F purifier. **b** Digital photographs show melamine foam, poly(N-isopropylacrylamide) hydrogel (PN), PN anchored on melamine foam skeleton (PN-F), and PN-F coated with N-isopropylacrylamide modified graphene (PG), respectively. The black PNPG-F is completely different from the transparent PN and white PN-F. **c** Cross-section SEM images of the hierarchical porous structure of PN. **d** SEM images of the highly stacked PG filter membrane coated outside. **e** Photographs show the wettability of PN and PN-F with (left) or without (right) solar heating (10 W cm$^{-2}$), and the corresponding water contact angle. **f** Water contact angle (WCA) cyclicity of PNPG, PN, and PNPG-F. **g** Plots show the weight recovery of various samples. Source data are provided as a Source Data file

by the storage modulus (G′) and the loss modulus (G″), respectively (Supplementary Figure 11). A wider linear viscoelastic region with a higher G′ value and G″ value could be found when either PG or melamine skeleton is added into PN. The increasing G′ and G″ values indicated that a tougher structure of PN is obtained by introducing PG nanosheets and a stronger skeleton[24]. As shown by the ultraviolet–vis near-infrared spectra in Supplementary Figure 12, the addition of PG and melamine contributed to the excellent absorption throughout the solar spectra with negligible optical loss.

**Water suction properties of PNPG-F**. Water suction procedure is a forward osmosis desalination process with PG as a semi-permeable membrane and polymer hydrogel as a draw agent. This draw agent must have a high osmotic pressure in the case of a forward osmosis, allowing saline water to pass across the semipermeable membrane[25]. Herein, PN was employed as the draw media of seawater and wastewater with a high concentration. PN is a 3D network hydrogel with hydrophilic functional groups or ionic groups that have attracted unwavering attention due to its ability of attracting greater amounts of water[26]. The cross-linked hydrogels exhibit a positive osmotic pressure of up to 4.2 MPa or 42 bar, which is higher than the osmotic pressure of seawater (~30 bar)[27]. The positive osmotic pressure inside PNPG-F will be further enhanced by counter ions that are balanced by the covalently incorporated ionic groups, as well as the introduction of a porous superhydrophilic melamine foam.

To investigate the water suction properties of PNPG-F, we first investigated the thickness effect of the PG membrane on the water suction rate of the PNPG-F purifier. As shown in Supplementary Figure 13, a series of circular PNPG hydrogels (60.0 mm in diameter) with various PG thickness were prepared. These membranes were irradiated under 4 suns to completely remove adsorbed water. These hydrogels were then immersed in pure water, NaCl aqueous solution (3.5 wt%), and methylene blue aqueous solution (MLB, 200 mg L$^{-1}$) at room temperature. The weight was monitored at different time intervals. It could be seen that as the thickness increased, the time needed for total weight recovery for a pure hydrogel rapidly increased from < 5 s to > 5000 s for a thickness from 0.5 mm to 5.0 mm (Supplementary Figure 13, Supplementary Movie 1), respectively.

The introduction of a superhydrophilic melamine skeleton significantly accelerates the swelling/deswelling rate of the PNPG-F purifier. On one hand, the PNPG-F purifier shows a superhydrophilic property below LCST owing to the porous structure of melamine foam and the low-energy surface of outstretched PNIPAm chains (Fig. 2d)[28]. The water contact angle (WCA) increased from 0° to 80° as the surface temperature increases from room temperature to above LCST (Fig. 2e and Supplementary Figure 14). PNPG-F showed an excellent reversible switch behavior between superhydrophilicity and hydrophobicity. In comparison with PNPG-F, PN was hydrophilic (the WCA was 31.3°) at room temperature and relatively hydrophobic above LCST (the WCA was 52.6°). On the other hand, the introduction of melamine skeleton avoided the bulk volume collapse of PNPG-F under sunlight irradiation (Supplementary Figure 14). Only ~10% of volume deduction was found in the case of PNPG-F. Pure PNPG hydrogel collapsed with a 55% decrease in volume. This smaller volume decrease and superhydrophilicity of PNPG-F contributed to a more rapid recovery than that of PNPG and PN, as shown in Fig. 2f. When the PNPG-F was immersed in aqueous solution, it quickly recovered to the original wet state within 60 s (Fig. 2f, Supplementary Movie 2 and Movie 3). This rapid water suction property remained even after 10 swelling/deswelling cycles.

**Transpiration and guttation under sunlight irradiation**. The mechanism underpinning transpiration and guttation is attributed to the photothermal effect of PG and a drastic water content change arises from the hydrophilicity/hydrophobicity switch of the PN, respectively. The thermoresponsiveness of PN enables the volume phase transition, which drives liquid water to be oozed for the guttation process[29]. The hydrophilicity/hydrophobicity switch of the PN was demonstrated by the water contact angle (WCA). The photograph in Fig. 3a showed the appearance of guttation of natural plant leaves. Remarkably, Fig. 3b displayed similar guttation of a PNPG-F purifier, mimicking the biological feature of natural plant leaves (Supplementary Figure 15).

The PG membrane acted as a solar absorber and a solar-heating convertor. Under 1-sun irradiation, the surface temperature of the wet PNPG-F purifier increased to an equilibrium temperature within 500 s (Fig. 3c). Typical infrared pictures of the surfaces indicated that the temperature reached ~55 °C under 1-sun irradiation (Supplementary Figure 16, and Supplementary Movie 4). The drastic change in the temperature of a sunlight-driven PNPG-F purifier enables the adsorbed water molecules to be simultaneously oozed, transpired under sunlight irradiation (Fig. 3d, e)[30]. It is obvious that the combined effect of transpiration and guttation endowed the PNPG-F with a much higher mass loss speed (4.2 kg m$^{-2}$ h$^{-1}$) than any other conventional sunlight evaporation devices reported so far (Fig. 3f)[31–38]. These results revealed that the PNPG-F can take full utilization of sunlight for realizing a significantly high rate for clean water production without extra fuel and electric energy. Details for calculation were shown in Supplementary Figure 17. Note that these experimental data of solar vapor generation are calibrated to dark-condition evaporation data[39].

The effect of water content on water collection was further investigated. The water content has less influence on the rate of transpiration (Supplementary Figure 18a). The water collection rate via transpiration slightly increases from 1.0 to 1.2 kg m$^{-2}$ h$^{-1}$ as the water content increases from 10 to 100%, which may be due to a larger channel between two adjacent GO sheets[40]. As shown in Supplementary Figure 18b, as the water content increases from 10 to 100%, the water collection rate (whole water collected from both transpiration and guttation) first increases and then decreases. The maximum collection rate reaches ~6.0 kg m$^{-2}$ h$^{-1}$ under 1-sun irradiation. In addition, in order to demonstrate the reliability of water collection rate test, we real-time monitored mass loss in the PNPG-F purifier with the device shown above (Supplementary Figure 18c and 18d). Similar tendency was obtained by this calculation method. The increase may be due to a faster increase of the surface temperature than the purifier with less water content. While the water-transport channels closed, the water content further decreases, inhibiting the steam generation and water oozing. The reason for the tuneable porous structure of the PNPG-F arose from the hydrophobic/hydrophilic switch of PNIPAm chains and a changeable laminal distance of PG nanosheets[29,40].

Noteworthily, the ratio of transpiration and guttation water can be further tuned (Supplementary Figure 19). According to Fig. 3d, the ratio of transpiration and guttation in the case of the PNPG-F purifier are 25% and 75%, respectively. We present a strategy to regulate the internal gap of PN and introduce a molecular mesh (Supplementary Figure 20) by premixture of oxidized graphene or polyvinyl alcohol (PVA). Oxidized graphene plays a role in regulating the internal gap of PN. Increasing the amount of oxidized graphene resulted in a smaller pore size of PN, which inhibits vapor generation. Consequently, the ratio of transpiration decreases from 25 to 11% as the concentration of oxidized graphene increases to 4.0 mg mL$^{-1}$ (Supplementary Figure 21). Furthermore, PVA in PN introduces molecular meshes to

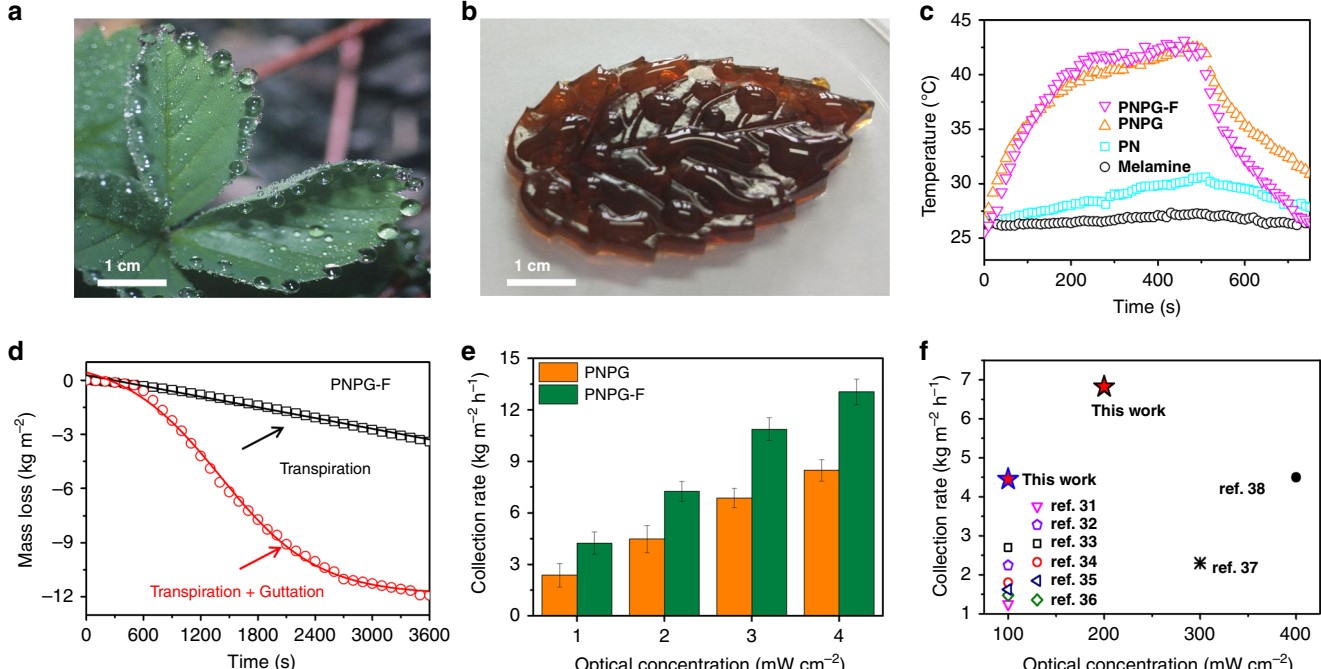

**Fig. 3** Appearance and water harvesting performance of the PNPG-F purifier. **a** Optical image of guttation of natural plant leaves. **b** Guttation phenomenon of the PNPG-F purifier. **c** Plots show the surface temperature of various samples under 1-sun irradiation at various time points. **d** The mass change of water with PNPG-F under 1-sun irradiation. **e** The water collection rate of PNPG and PNPG-F under various optical concentrations. Error bars were determined based on the standard deviation measured multiple times. **f** Comparison of water harvesting performance of the PNPG-F purifier and the previously reported word under various optical energy densities. Source data are provided as a Source Data file

establish a hierarchical water transport channel, enhancing transpiration[41]. As shown in Supplementary Figure 22, after removing PVA by dialysis, macro/micropores were constructed in the cell walls of PN. This porous structure facilitated the vapor generation. Consequently, the ratio of transpiration increases from 60 to 80%.

**Mechanism for high-efficiency solar transpiration and guttation.** The high rate of sunlight-driven transpiration and guttation resulted from the photothermal properties of the PG layer and the unusual negative temperature-response behavior of PN, respectively. Under constant solar illumination, the temperature reached ~55 °C and continuous steam generation occurred (transpiration). Simultaneously, guttation arises from the drastically altered water content of a thermoresponsive PN, which could be significantly affected by small changes in temperature because their polymer chains switch from hydrophilic to hydrophobic above LCST. When PNIPAm chains were anchored to melamine skeleton, the synergistic effect of osmotic pressure of PN and capillary force driven by melamine skeleton endowed the PNPG-F purifier an enhanced capability of water suction[42].

We then established a simple PNIPAm molecular model capable of reproducing the essentials of the experimentally observed coil–globule scenario of PN in aqueous solution (Supplementary Figure 23). Two different models were established to simulate the PN and PN-F systems, respectively (Supplementary simulation method). MD simulation results displayed that PNIPAm chains for PN are hydrated and surrounded by water molecules with a clustering configuration due to the coil-like PNIPAm molecules at a temperature of 280 K below the LCST (as shown in Supplementary Figure 24). The clustering configuration further collapsed with dehydration when heated to 340 K above LCST, which is consistent with previous simulation results[43,44]. However, by fixing the chain ends of

PNIPAm molecules to mimic the anchoring effect on the rigid melamine skeleton for PN-F, the chain clustering was largely suppressed below LCST, leading to the exposure of more hydrophilic amide groups for improving water suction below LCST, and the significant reduction in the number of HBs above LCST as discussed below.

Figure 4c showed that both PN and PN-F above LCST contain less PNIPAM–water hydrogen bonds (HBs) (Supplementary simulation method) than those below LCST, which was attributed to the collapsed structure and dehydration of PNIPAm. When increasing the temperature from 280 to 340 K, the HB reduction of the PN-F system is 35.71% larger than that of the PN system. It has been revealed that the water swelling/deswelling capability of PNIPAm is greatly relevant to its change in HBs below and above LCST[45,46]. A larger change in HBs resulted in a faster water suction and oozing speed of the PNPG-F purifier.

In order to investigate the dehydrated behavior, we further confirmed the dehydration of collapsed PNIPAm chains through the water-accessible surface area analysis. Compared to the PN system, the total interfacial area between anchored PNIPAm chains and water in the PN-F system is larger at 280 K (Fig. 4d), because the anchored structure suppresses the chain collapse and clustering, resulting in much more hydration. During further equilibration at 340 K, the surface area of PNIPAm occupied by water in two systems decreased gradually. A larger surface area drop was found for PN-F (8128.1 Å$^2$) than that of PN (4010.5 Å$^2$). By examining the change in the density of O (water) and C (amide group) as shown in Supplementary Figure 25, we observed that the anchored PNIPAm with a more available surface area favored hydration and dehydration below and above LCST, respectively. Therefore, the simulation suggests that the anchoring effect of the PNIPAm chains onto the melamine skeleton in the PN-F system could prevent the molecules from clustering, which exposed more hydrophilic amide groups and retained a

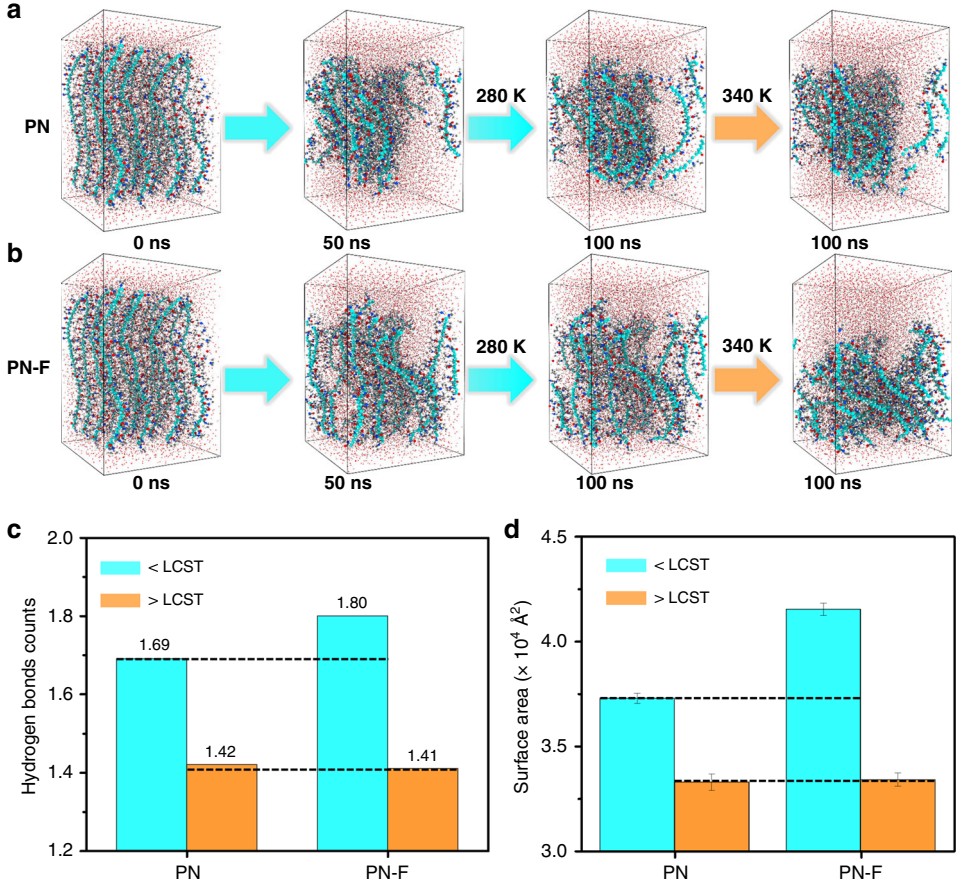

**Fig. 4** Simulation results of the coil–globule behavior of PN and PN-F in water environment. **a, b** Snapshots of PN and PN-F, after equilibration at 280 K over 100 ns and heating at 340 K over another 100 ns. Both PN and PN-F systems consist of 16 PNIPAm chains and 7680 water molecules: hydrogen atoms are depicted in white, oxygen atoms in red, nitrogen atoms in deep blue, backbone carbon atoms in light blue, and other carbon in gray. **c** Normalized number of hydrogen bonds in completely equilibrated PN and PN-F systems at 280 and 340 K. **d** Total water-accessible surface area of PN and PN-F at 280 and 340 K. Error bars reflect the standard deviation in surface area during the last 10 ns. Source data are provided as a Source Data file

more pore-like structure below LCST, which enhanced the water suction, and the significant reduction in the number of HBs above LCST could enhance the water oozing property. The combination of these two factors accelerates the swelling/deswelling of the PNPG-F purifier.

**Separation performance of the PNPG-F purifier powered by sunlight.** To validate the feasible application of our sunlight-driven purifier in water purification, we conducted ion and molecule separation experiments to test the water purification performance. Figure 5 summarized the rejection results for various ions and molecules via the PNPG-F purifier powered by sunlight. The results of electrolyte salts rejection (Fig. 5a) indicated that the ionic rejection decreased with concentration and increased with the valence ratio, $Z^-/Z^+$ (Supplementary Figure 26), which is consistent with the previous results of carbon nanotubes[47], graphene oxide[48], and molybdenum disulfide membranes[49], which can be explained by Donnan exclusion theory[50]. The negative zeta potential results further gave an explanation of the ionic rejection performance (Fig. 5b). When $Z^-/Z^+ = 0.5$, the rejection of $MgCl_2$ and $CaCl_2$ is nearly the same within an allowable error. $CuCl_2$ is much higher than that even close to the rejection when $Z^-/Z^+ = 1.0$ in our experiments. This is because $Cu^{2+}$ ions tend to coordinate with functional groups on the edge of PG[51], leading to a higher permeate barrier, along with a much larger hydration radius than $Na^+$ and $K^+$[52].

The rejection of dye molecules through the PNPG-F purifier was dominated by the size-sieving mechanism showing large rejection for the large molecules, and the rejection rate decreases with concentration. The X-ray diffraction (XRD) peak of PG membranes was 4.5°, while the diffraction peak shifted to the right to 5.7° under sunlight irradiation. Correspondingly, the interlayer spacing ($d$) decreases from 18.7 to 14. 4 Å, respectively. The smaller distance under sunlight irradiation arises from the shrinking of PNIPAm chains. PG membrane soaked in water showed $d \approx 13.3$ Å due to the intercalation of 2–3 layers of water molecules[53]. For the molecules with the same charge, the rejection mechanisms are beyond the size sieving. For example, the size of Rhodamine B (RB) and Rhodamine 6G (R6G) is close, while the rejection of the former is smaller; the size of methyl orange (MO) is smaller than methyl blue (MB) while the rejection of the latter is larger. Because the PG is negatively charged, electrostatic interactions can influence the mobility of dyes, resulting in a higher rejection rate of negatively charged dyes than positively charged ones (Supplementary Table S1).

**Laboratory and outdoor solar desalination.** For practical application, we further constructed a waterwheel-like structure (Fig. 6a–c). The waterwheel blades support the continuous immersion and sunlight-driven water collection at a speed of 1 revolution every 10 min. This device offered a practical utilization of our PNPG-F purifier. We performed cycle tests of rejection of

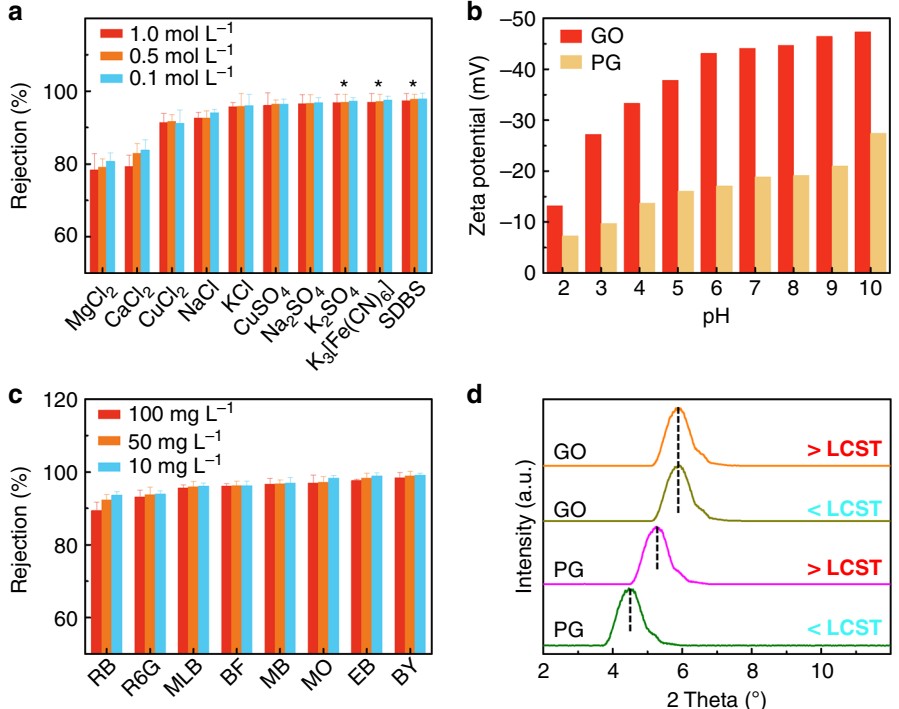

**Fig. 5** Performance of PNPG-F for single ion and molecule separation. **a** Rejection of ions through PNPG-F under sunlight irradiation. The asterisk (*) denotes the concentration of sodium dodecyl benzenesulfonate (SDBS, 0.05, 0.01, and 0.005 M), $K_3[Fe(CN)_6]$ (0.1, 0.05, and 0.001 M), and $K_2SO_4$ (0.1, 0.05, and 0.001 M), respectively. **b** The zeta potentials of graphene oxide (GO) and pure PG dispersion measured in a pH range of 2–10. **c** Rejection of dye molecules through PNPG-F under sunlight irradiation. **d** X-ray diffraction (XRD) peak showing shifts of the (001) peak due to the shrinking of PNIPAm chains under sunlight irradiation. Error bars denote standard deviations from three samples measured multiple times. Source data are provided as a Source Data file

$K_3[Fe(CN)_6]$ and RB. The cycle tests demonstrated that ion and molecule rejection rate remained after 100 cycles of repeated suction and guttation (Supplementary Figure 27). Furthermore, the salt durability of the as-prepared PNPG-F purifier is demonstrated. The PNPG-F purifier presents a stable water collection rate and ionic/molecular rejection ratio on exposure to the seawater for 30 days. The results of this solar-driven water collection test demonstrated the durability of the PNPG-F purifier (Fig. 6d).

To evaluate the feasible performance of the sunlight-driven purifier, a 8.0-cm × 4.0-cm × 0.5-cm PNPG-F was prepared with one side absent of the PNPG coating (Supplementary Figure 28) to realize the lake water and sea water purification. The PNPG-F purifier was placed in a sealed glass funnel with the uncoated side upside-down (Supplementary Movie 5). The purification of lake water of the Tsinghua University was carried out in the lab. The experiments of the Bohai Sea (average salinity 3.0%) water purification were conducted under natural sunlight from 7:00 to 19:00 (Fig. 6e). After desalination, the salinities of all the brine samples were significantly decreased to below 0.7 wt% in guttation water, and 0.002 wt% in transpiration water (Supplementary Figure 29). A daily yield of 21.1 kg m$^{-2}$ was recorded under natural sunlight irradiation from seawater.

## Discussion

Inspired by natural plant leaves, this study demonstrated a self-propelled purifier based on a thermal-responsive hydrogel (PN) and solar-heating filtration membrane (PG). The advantageous cooperation of a self-propelled water suction, the exceptional molecular rejection, and sunlight-driven transpiration and guttation endows the as-prepared PNPG-F purifier a capability of rapid and reversible clean water production. We specifically

combined transpiration and guttation effect, to the best of our knowledge, resulting in a record-high water collection rate of up to 4.2 kg m$^{-2}$ h$^{-1}$ under 1-sun irradiation. The PNPG-F purifier we developed with the synergistic effect of the forward osmosis/membrane desalination and transpiration could not only largely enhance the water collection rate, but also worked well in anti-fouling, giving another solution to realizing the practical usage of solar-powered clean water production.

Noteworthily, the process of water suction starts from the permeation of the feed to the draw media (PN-F hydrogel) across the semipermeable membrane (PG layer). The draw hydrogel is subsequently recovered by separating clean water under solar irradiation. The procedure can be repeated with a waterwheel-like device shown in Fig. 6 for continuous water production. Obviously, this forward osmotic desalination procedure avoids the clog of a water transport channel by prohibiting salt precipitation on the surface of photothermal materials. Salt precipitation is the main contamination for a conventional solar steam generation system, which will significantly decrease light utilization and reduce water transport efficiency. That repeated immersion procedure endows our design the ability of having a lower/reversible membrane fouling, ensuring a durable distillation performance. In addition, the procedure is negligibly affected by a variety of contaminants in the feed solution.

In convention, water molecules should be heated from a liquid state to a gaseous state. Subsequently, liquid-state water has to be produced by further condensing. Therefore, the solar steam system requires high energy consumption to evaporate water and condense it for collection. The plant leaves inspired design is different from conventional solar steam generation systems due to cooperating transpiration and guttation. In this case, water collection (through guttation) involves only two steps that are liquid absorption and regeneration of clean liquid water, and

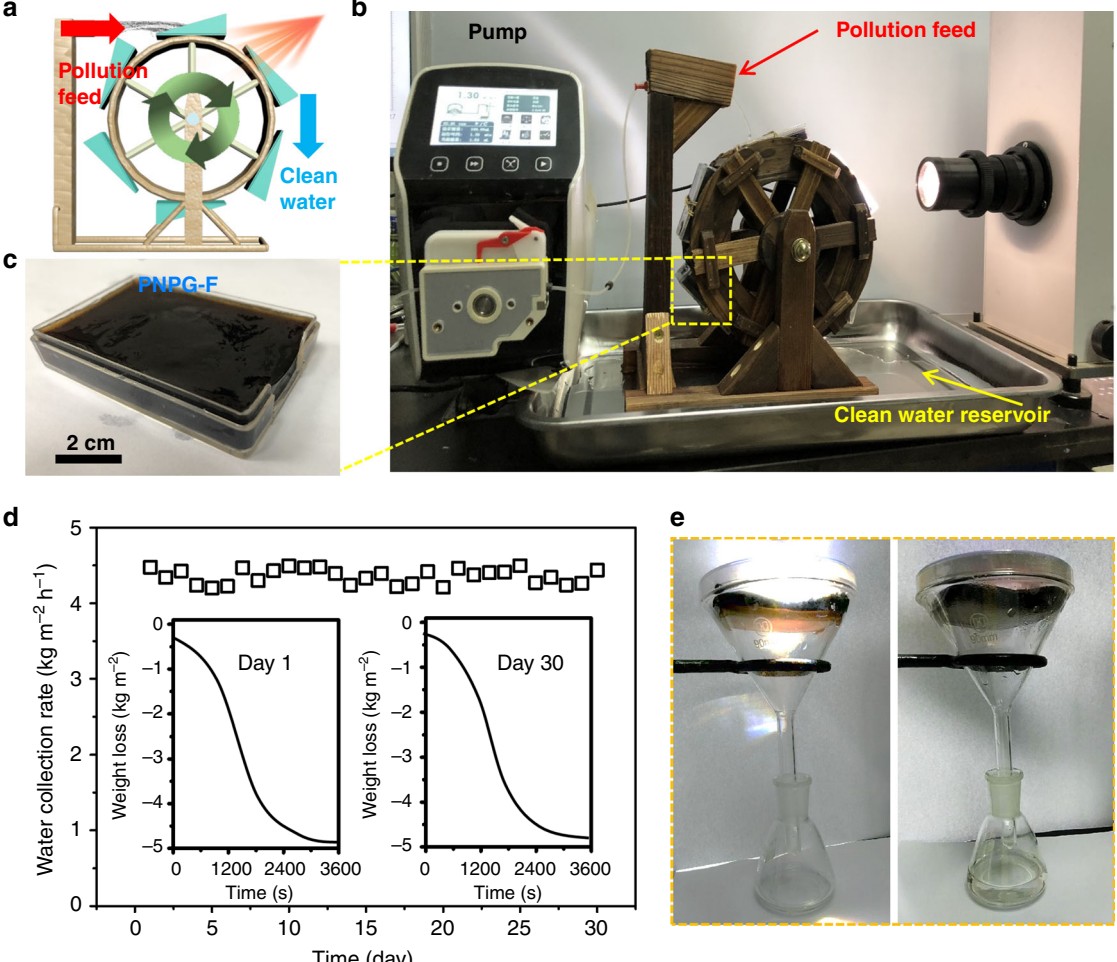

**Fig. 6** Continuous water purification using PNPG-F. **a** Schematic of a water collector based on a waterwheel structure. **b** The PNPG-F purifier is fixed on the gear of the waterwheel. Polluted water is fed by a pump. As the waterwheel rotates at a speed of 1 revolution every 10 min, clean water is continuously collected under solar irradiation. **c** A digital image of a single PNPG-F. **d** Water collection rate of the PNPG-F purifier immersed in seawater for 30 days. The results of the collection demonstrate the durability of the PNPG-F purifier. Insets are the typical water collection behavior on exposure to seawater for 1 and 30 days. **e** A simple device for the whole collection of water from transpiration and guttation

simultaneously, the regeneration of the absorbents. The transpiration is similar to a conventional solar steam generator, which produces a rate of ca. 1.1 kg m$^{-2}$ h$^{-1}$ under 1-sun irradiation. On the other hand, the liquid to liquid collection rate (guttation) is ca. 3.1 kg m$^{-2}$ h$^{-1}$ under 1-sun irradiation. Water could be collected directly in the liquid state without further condensing. The liquid water could be produced at a relatively low temperature (~31 °C). Simultaneously, solar-heating steam could also be collected. The activation energy for water release below the LCST is 95.3 kJ mol$^{-1}$, which is similar to that for the evaporation of water. However, the activation energy for water release above the LCST decreases to as low as 3.8 kJ mol$^{-1}$, which is much lower than that below the LCST[30]. These significantly lower energy consumptions promote water collection efficiency. Meanwhile, by rational combination with waterwheel-like devices, PNPG-F was capable to produce freshwater from seawater with a daily yield of 21.1 kg m$^{-2}$ under natural sunlight, which is enough for daily use to several people. This work provides another route to take full use of solar energy for high-rate water purification and production.

## Methods
**Synthesis of GO.** GO nanosheets were prepared according to our previous report[1]. The resultant GO dispersion was centrifuged several times to remove nonexfoliated

graphite particles, followed by being dialyzed for at least 1 week, when the ionic conductivity of the dialysis water was lower than 5 μm cm$^{-1}$. The GO nanosheets have lateral sizes of 20–30 μm.

**Preparation of PNPG-F.** The preparation of the hydrogel follows the procedure reported elsewhere[2], specifically, 10 mL of aqueous solution containing 10 mg of APS and 10 μL of TEMAB was sprayed into a cuboid (5.5 × 3.0 × 0.5 cm) melamine foam by an atomizer, followed by being dried at room temperature. In total, 283 mg of NIPAm and 12.5 mg of MBAAm were dissolved in 5 mL of water that is rea-erated by nitrogen for at least 30 min. The above melamine foam was immersed into this solution and sealed at room temperature overnight. The as-prepared products were abbreviated as PN-F. Meanwhile, the same PNIPAm aqueous solution was prepared, into which 5 mg of GO was added and sonicated for 5 min for homogeneous dispersion. The PN-F was then coated with this NIPAm GO solution, and was further annealed at 50 °C overnight, which is noted as PNPG-F. The products were finally dialyzed in Ultrapure Milli-Q water for at least 1 week to remove any unreacted molecules.

**Calculation of water collection efficiency.** The experiments were conducted at an ambient temperature of ~23 °C and a humidity of ~65%. The water collection experiments were performed under the illumination of a solar simulator (CEL-HXF300). As soon as turning up the light, the mass change was immediately tracked in real time by an electronic mass balance communicated to a computer. In the case of guttation efficiency, the oozed water was drained through a channel (Supplementary Figure 17a). The whole weight loss over the entire process arose from both transpiration and guttation. As shown in Supplementary Figure 17b, in reference to the transpiration efficiency, the samples were floated in a beaker. The oozed water was recollected by the water container. The weight loss arose only

from solar-heating evaporation. The whole weight loss over the entire process arose from both transpiration and guttation. The evaporation rates in the dark field were subtracted from all of the measured evaporation samples.

In order to avoid the influence of shape change and surface determination on the calculation result, we prepared a series of sound- shaped samples with a diameter of 90 mm (the diameter of the sunlight spot is 60 mm, Supplementary Figure 17c). The effective illumination area only depends on the sunlight spot, avoiding the effect of size change of samples on the calculation. Only the upper surface samples were coated with PG to avoid the heating of the side surface (Supplementary Figure 17d). We have added the above details in the revised manuscript.

**Characterization**. The porous structure and surface topography of melamine before and after PN growth was observed by using a Sirion-200 scanning electron microscope (FEI, USA). The absorbance spectra of the purifier were measured using a Varian UV–vis spectrophotometer (Cary 5000, USA), coupled with an Agilent integrating sphere. An IR camera (Fluke) was used to measure the temperature increase under solar irradiation. The wettability change of the purifier after solar irradiation was measured by the contact angle analysis system with a 3.0-μL water droplet. Raman spectra were obtained by using LabRAM HR Evolution (HORIBA Jobin Yvon, France) Raman microscope with a 514-nm laser. Elemental analysis was finished using XPS spectra taken out by an ESCALAB 250XI photoelectron spectrometer (ThermoFisher Scientific, USA). X-ray diffractions (XRD) were carried out using a D8 Advance X-ray diffractometer with Cu Kα radiation ($\lambda = 0.15418$ nm, Bruker, Germany). The transpiration and guttation experiments were conducted in the lab using a solar simulator with an optical filter for the standard AM 1.5G spectrum. The optical concentrations of 1 sun and 2 sun are 100 and 200 mW cm$^{-2}$, respectively. The contact angle was measured using a Dataphysics OCA 15pro CA measuring instrument (DataPhysics Instruments GHPH, Filderstadt).

## Data availability
The data presented within this paper are available from the corresponding author upon reasonable request.

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

## Acknowledgements

We acknowledge financial support from the China Postdoctoral Science Foundation Grant (Grant No. 2017M620037, 2018T11008), the National Key R&D Program of China (2016YFA0200200, 2017YFB1104300), and the Natural Science Foundation of China (51433005, 21674056, 51673108, and 51673026), NSFC-MAECI (51861135202). We sincerely thank Ke Zhou from Tsinghua University for his insightful comments on the molecular and ionic rejection results. Computations were carried out on the "Explorer 100" cluster system of Tsinghua National Laboratory for Information Science and Technology.

## Author contributions

L.Q. supervised the entire project. L.Q. and H.G. conceived the idea and co-wrote the paper. H. Geng performed the whole material preparation and characterization and carried out data analyses. Q.X. and T.M. performed the molecular dynamics simulations. M.W., H.M., P.Z., T.G., and C.L. gave advice on the experiments and reviewed the paper. All the authors discussed the results and commented on the paper.

## Additional information

**Competing interests:** The authors declare no competing interests.

