## [Peer Review File · Nature Communications]

Reviewers' comments:

Reviewer #2 (Remarks to the Author):

The authors tried to carefully respond the previous reviewers' comments. It seems now ready for publication.

Reviewer #3 (Remarks to the Author):

I have the manuscript "Plant Leaves Inspired Sunlight-Driving Purifier for High-Efficiency Clean Water Production" and the authors' response for Nature Nanotechnology, which appears to me very thorough and comprehensive.

The authors have clearly convinced the previous reviewers the novelty of the work and it should be published in Nature Communications without any further delays.

Reviewer #4 (Remarks to the Author):

In this manuscript, authors propose a sunlight-driving purifier for water purification and production. The idea is to use (i) a negative temperature response hydrogel, anchored onto a superhydrophilic melamine skeleton, and (ii) a layer of modified graphene filter membrane coated outside. After

being irradiated by sunlight, the device releases clean water by transpiration (liquid-to-vapor process) and guttation (liquid-to-liquid process).

The idea to use a liquid-to-liquid purification process by a coated hydrogel is interesting. However, before assessing the suitability of this paper for publication, the following issues must be properly discussed and clarified.

1) A liquid-to-liquid purification process implies that the modified graphene filter membrane (coated outside the hydrogel) works as a partially permeable membrane to remove ions and contaminant molecules. During the water suction step, (almost) pure water is supposed to enter into the hydrogel and this will be later released by transpiration and guttation, under sunlight irradiation. However the above scenario requires that a pressure different overcoming the osmotic pressure of seawater is applied across the modified graphene filter membrane during suction, because of the capillarity effect due to the hydrophilicity of the hydrogel. The problem is that the ideal osmotic pressure of seawater is roughly 30 bar and it seems unlikely that the latter can be overcome by the capillarity effect only (30 bar corresponds to 300 m of water column). In real industrial processes, overcoming the osmotic pressure by reverse osmosis requires to pressurize one side of the partially permeable membrane up to 60-70 bar. Are the authors really sure that the capillarity effect driven by the hydrophilicity of their material can suck with a strength corresponding to (at least) 30 bar? Otherwise, salt ions and pollution molecules would be (at best) adsorbed on their contaminated material...

2) The authors claim that their device has advantages with regards to photothermal materials, where the salt deposition could not be avoided effectively. Actually, where does the salt go in the proposed device? I assume that the salt will deposit on the external side of the modified graphene filter membrane coated outside the hydrogel. How will the salt be removed? For example, in the waterwheel-like structure, showed in Fig. 6, where does the salt go and how it is removed?

3) The authors say that guttation, namely the liquid-to-liquid purification process, can be explained by (a) an unusual negative temperature-response behavior of the hydrogel and (b) a drastic phase change, which arises from the hydrophilicity/hydrophobicity switch of the hydrogel. Are both effects relevant with the same magnitude? Which one is prevailing and why? The authors should prove that these effects can *quantitatively* explain the observed guttation, e.g. in terms of changes in the materials volume.

4) In a former paper by some of the current authors, namely Zhao F. et al. "Highly efficient solar vapour generation via hierarchically nanostructured gels" in Nat. Nanotech. 13, 489-495 (2018), they have already presented a liquid-to-liquid purification process, based on a claimed "reduction in

vaporization enthalpy". In the above paper, some of the current authors claimed that the "reduction in vaporization enthalpy" could be explained by the water cluster theory. That "reduction in vaporization enthalpy" and its explanation raised many doubts in the scientific community dealing with solar desalination. Hence, the authors may take this opportunity for clarifying this situation and for explaining why the present liquid-to-liquid purification process based on guttation is different/better.

5) The authors claim a purification rate of $4.2 \text{ kg m}^{-2} \text{ h}^{-1}$. I am not convinced about how this rate is computed. Reading the manuscript, I think that this is the release rate of purified water from the proposed material under sunlight, but the latter has nothing to do with the (continuous) purification rate. In fact, the proposed device cannot continuously operate to generate clean water with a purification rate of $4.2 \text{ kg m}^{-2} \text{ h}^{-1}$. This is clear by looking at Fig. 6, where the waterwheel-like structure is shown: Clearly a big rotating structure is needed for allowing the proposed material to operate. Hence much larger surface areas are needed for installation and this reduces significantly the continuous purification rate per unit of surface area. Please correct this point in your abstract for avoiding misleading comparisons with other devices, which are designed for real continuous operation.

6) The authors claim that "As long as the water collection behaviour is the condensation of steam, the efficiency is pinned near the efficiency limits (ca. $1.5 \text{ kg m}^{-2} \text{ h}^{-1}$)". This is not true. The condensation energy can be fruitfully re-used multiple times, as far as the corresponding temperature is above the ambient one. See the paper by Eliodoro Chiavazzo et al., *Nature Sustainability*, vol 1, pages 763–772 (2018), <https://doi.org/10.1038/s41893-018-0186-x>.

7) There are some relevant papers missing in the list of references, e.g. the pioneering paper by Hadi Ghasemi et al., *Nature Communications*, vol 5, 4449 (2014), <https://doi.org/10.1038/ncomms5449>. Maybe the authors could refer to some recent review papers, e.g. Chen et al., "Challenges and Opportunities for Solar Evaporation", *Joule* (2018), <https://doi.org/10.1016/j.joule.2018.12.023> or Wang, P., "Emerging investigator series: the rise of nano-enabled photothermal materials for water evaporation and clean water production by sunlight," *Environmental Science: Nano*, Vol. 5, 1078, 2018, <http://doi.org/10.1039/C8EN00156A>.

Minor:

The authors say that "According to Fig. 3d, the ratio of transpiration and guttation in case of PNP-G-F purifier are 75% and 25%, respectively". Isn't the other way around? I mean guttation is 75%, right?

Reviewers' comments:

Reviewer #2 (Remarks to the Author):

The authors tried to carefully respond the previous reviewers' comments. It seems now ready for publication.

Reviewer #3 (Remarks to the Author):

I have the manuscript “Plant Leaves Inspired Sunlight-Driving Purifier for High-Efficiency Clean Water Production” and the authors' response for Nature Nanotechnology, which appears to me very thorough and comprehensive.

The authors have clearly convinced the previous reviewers the novelty of the work and it should be published in Nature Communications without any further delays.

Reviewer #4 (Remarks to the Author):

In this manuscript, authors propose a sunlight-driving purifier for water purification and production. The idea is to use (i) a negative temperature response hydrogel, anchored onto a superhydrophilic melamine skeleton, and (ii) a layer of modified graphene filter membrane coated outside. After being irradiated by sunlight, the device releases clean water by transpiration (liquid-to-vapor process) and guttation (liquid-to-liquid process).

The idea to use a liquid-to-liquid purification process by a coated hydrogel is interesting. However, before assessing the suitability of this paper for publication, the following issues must be properly discussed and clarified.

1) A liquid-to-liquid purification process implies that the modified graphene filter membrane (coated outside the hydrogel) works as a partially permeable membrane to

remove ions and contaminant molecules. During the water suction step, (almost) pure water is supposed to enter into the hydrogel and this will be later released by transpiration and guttation, under sunlight irradiation. However the above scenario requires that a pressure different overcoming the osmotic pressure of seawater is applied across the modified graphene filter membrane during suction, because of the capillarity effect due to the hydrophilicity of the hydrogel. The problem is that the ideal osmotic pressure of seawater is roughly 30 bar and it seems unlikely that the latter can be overcome by the capillarity effect only (30 bar corresponds to 300 m of water column). In real industrial processes, overcoming the osmotic pressure by reverse osmosis requires to pressurize one side of the partially permeable membrane up to 60-70 bar. Are the authors really sure that the capillarity effect driven by the hydrophilicity of their material can suck with a strength corresponding to (at least) 30 bar? Otherwise, salt ions and pollution molecules would be (at best) adsorbed on their contaminated material...

Reply: We greatly thank for these insightful comments. As reminded by the referee, draw agents must have a high osmotic pressure in case of forward osmosis, allowing saline water passes across the semipermeable membrane. Herein we employed polymer hydrogel as the draw media of seawater or wastewater with high concentration. The enough osmotic pressure of polymer hydrogel for water suction has been previously demonstrated [For example, M. Qasim, *et al. Desalination* **2017**, 423, 12–20; D. Li, *et al. Soft. Matt.* **2011**, 7, 10048–10056; D. Qin, *et al. Environ. Sci. Technol. Lett.* **2018**, 52, 1421–1428; M. Elimelech, *et al. Science* **2011**, 333, 712–717; L. Arens, *et al. Macromol. Chem. Phys.* **2017**, 218, 1700237, and so on]. Besides, PNIPAm hydrogel we chose is 3D network of polymer chains that are cross-linked by chemical bonds. It is a superabsorbent among various hydrogels that have attracted unwavering attention due to its ability of water suction [P. Schexnailder, *et al. Colloid Polym. Sci.* **2009**, 287, 1–11; A. Okafuji, *et al. Macromol. Rapid Commun.* **2016**, 37, 1130–1134]. 3D structure of PNPG-F with hydrophilic functional groups or ionic groups on the comonomer unit are capable of attracting greater amounts of water. The positive osmotic pressure inside PNPG-F will be further enhanced by counter ions that are balanced by the

covalently-incorporated ionic groups. The crosslinked hydrogels exhibit a positive osmotic pressure of up to 4.23 MPa or 42 bar, which is much higher than 30 bar [D. Li, *et al. Chem. Commun.* **2011**, 47, 1710–1712; H. Wack, *et al. Polymer* **2009**, 50, 2075-2080].

In particular, the osmotic pressure could be substantially increased by structure engineering. In the current case, we introduced another porous superhydrophilic melamine foam into PNIPAm hydrogel, which will no doubt further increase the osmotic pressure due to the ionic groups and porous characteristics. This was demonstrated experimentally and theoretically. Experimentally, the introduction of a superhydrophilic melamine foam endowed the PNIPAm hydrogel a superhydrophilicity, which contributes to a more rapid recovery than that of PNPG and PN. When the PNPG-F was immersed in aqueous solution, it quickly recovered to the original wet state within 60 s (Figure 2f, Supplementary Movie 2 and Movie 3). Theoretically, a simple PNIPAm molecular model was established to reproduce the essentials of the experimentally observed coil-globule-coil scenario of PN in aqueous solution (Figure 4, Supplementary Figure S22-26). A larger change in hydrogen bonds resulted in a faster water suction and oozing speed of PNPG-F purifier (Figure 4).

The related discussion has been included in the revised manuscript. See pages 5 and 6, a new paragraph was added, which was highlighted by yellow.

2) The authors claim that their device has advantages with regards to photothermal materials, where the salt deposition could not be avoided effectively. Actually, where does the salt go in the proposed device? I assume that the salt will deposit on the external side of the modified graphene filter membrane coated outside the hydrogel. How will the salt be removed? For example, in the waterwheel-like structure, showed in Fig. 6, where does the salt go and how it is removed?

Reply: Thanks for the good questions. That is a technical issue, which is commonly

come across with in any case of membrane separation procedure. The outside that is contacted with feed solution should be cleaned first before production of clean water and must be cleaned after ionic/molecular separation procedure for next utilization. Particularly, with rational design we are able to skip the clean step. As shown in Supplementary Figure S28, we fabricated a single PNPG-F purifier with one side being coated with PN while left another side uncoating. Pollute feed is contacted with coating side, while clean water is collected through the uncoated face. Alternatively, for case of waterwheel like device, after being exposed to pollution, we further cleaned it before solar irradiation. Compared with solar steam generation system, the clean process is much easier via a little amount of clean water. That is similar with conventional membrane separation procedure.

In fact, forward osmosis has been considered as an energy-efficient and economical alternative to the conventional seawater desalination technologies without considering the fouling of the membrane [see ref. 10, a review paper in our manuscript; Also, M. Elimelech, *et al. Science* **2011**, 333, 712–717; S. Zhao, *et al. J. Membr. Sci.* **2012**, 396, 1–21; G. Blandin, *et al. Desalination* **2015**, 363, 26–36; R. V. Linares, *et al. Water Res.* **2016**, 88, 225-234, and so on]. As shown in our manuscript, the process of forward osmosis desalination starts from the permeation of the feed to the draw media (PN-F hydrogel) across the semipermeable membrane (PG layer). The draw hydrogel is subsequently recovered by separating clean water under solar irradiation. Obviously, this forward osmosis desalination procedure avoids the clog of water transport channel by prohibiting salt precipitation on the surface of photothermal materials. The procedure is repeated with a waterwheel like device shown in Fig. 6 for continuous water production. Salt precipitation is the main contamination for conventional solar steam generation system which will significantly decrease light utilization and reduce water transport efficiency. That repeated immersion procedure endows our design the ability of having lower/reversible membrane fouling, ensuring durable distillation performance. In addition, the procedure is negligibly affected by a variety of contaminants in the feed solution.

We further discussed those advantages in the revised manuscript. See details in page 16, which is highlighted by yellow.

3) The authors say that guttation, namely the liquid-to-liquid purification process, can be explained by (a) an unusual negative temperature-response behavior of the hydrogel and (b) a drastic phase change, which arises from the hydrophilicity/hydrophobicity switch of the hydrogel. Are both effects relevant with the same magnitude? Which one is prevailing and why? The authors should prove that these effects can *quantitatively* explain the observed guttation, e.g. in terms of changes in the materials volume.

Reply: Many thanks for these professional comments. In fact, the “temperature-response behavior” is the property of the hydrogel, the “a drastic phase change” is the result of temperature response, and the “hydrophilicity/hydrophobicity switch of the hydrogel” is the essential mechanism. Guttation arises from the drastically altered water content of thermo-responsive PNIPAm hydrogel, which is could be significantly affected by small changes in temperature because their polymer chains switch from hydrophilic to hydrophobic above LCST. This is the essential mechanism for guttation. We have united the explanation of guttation into the drastic water content change of PNPG-F due to the hydrophilicity/hydrophobicity switch of the hydrogel in the revised manuscript.

It is true that guttation could be quantitatively explained and reflected by the hydrophilicity/hydrophobicity switch of the hydrogel. As we have demonstrated, parameters including solar concentration (Fig. 2e), thickness (Supplementary Fig. S13), water content (Supplementary Fig. S18), and the introduction of PVA and GO (Supplementary Fig. S20 and S21) influencing hydrophilicity/hydrophobicity switch have great effect on the water collection behavior. For your convenience we described one of the quantitative demonstrations below. We real-time monitored the water collection behavior and surface temperature increase under one sun irradiation of

PNPG-F under sunlight irradiation. As shown in Fig. 2, as soon as the temperature of the PNPG-F increase to ~ 32 °C, the switch of hydrophilic to hydrophobic occurs, initiating guttation and significantly accelerating water collection rate.

We further classified the mechanism for guttation as suggested in the revised manuscript. In page 8, “phase change” had been changed to “drastic water content change” page 10, 3 sentences were added.

4) In a former paper by some of the current authors, namely Zhao F. et al. “Highly efficient solar vapour generation via hierarchically nanostructured gels” in Nat. Nanotech. 13, 489-495 (2018), they have already presented a liquid-to-liquid purification process, based on a claimed “reduction in vaporization enthalpy”. In the above paper, some of the current authors claimed that the “reduction in vaporization enthalpy” could be explained by the water cluster theory. That “reduction in vaporization enthalpy” and its explanation raised many doubts in the scientific community dealing with solar desalination. Hence, the authors may take this opportunity for clarifying this situation and for explaining why the present liquid-to-liquid purification process based on guttation is different/better.

Reply: Thanks a lot for your good suggestion. In the mentioned report, we made efforts to elevate the liquid-to-vapor water evaporation rate via precisely engineering the nanostructure of a polyvinyl alcohol and polypyrrole hydrogel. Consequently, water content in the hydrogel could be regulated, as well as an expedited water absorption by the hierarchical pathways could be realized. The solar steam generation rate is thus significantly increased to $3.2 \text{ kg m}^{-2} \text{ h}^{-1}$, which represents the highest number at that moment.

However, in this work, the PNPG-F purifier offers a brand-new water collection manner, *e.g.* the liquid-to-liquid collection behavior. That is quite different from conventional solar steam generation system, where water molecules need to be heated from a liquid state to a gaseous state. Subsequently, liquid state water has to be produced by further condensing. In this case, water collection (through guttation)

involves only two steps, i.e. liquid absorption, and regeneration of clean liquid water, simultaneously, the regeneration of the absorbents. The mechanism for guttation is attributed to the photothermal effect of PG and a drastic water content change arises from the hydrophilicity/hydrophobicity switch of the PNIPAm hydrogels, respectively. Under sunlight irradiation, the thermo-responsive of PNIPAm hydrogel enables the hydrophilicity/hydrophobicity switch, which drives liquid water to be oozed for the guttation process [Qu et al. *Angew. Chem. Int. Ed.* **2018**, 57, 15435–15440]. This liquid to liquid collection rate (guttation) is *ca.* $3.1 \text{ kg m}^{-2} \text{ h}^{-1}$ under one sun irradiation. That is due to a much lower activation energy and temperature ($32 \text{ }^{\circ}\text{C}$ is enough) for switch of hydrophilic/hydrophobic. The activation energy for water release below the LCST is 95.3 kJ mol^{-1} , which is similar to that for the evaporation of water. However, the activation energy for water release above the LCST decreases to as low as 3.8 kJ mol^{-1} , much lower than that below the LCST [Miyata *et al. Nat. Commun.* **2018**, 9, 2315]. More importantly, along with guttation, water transportation simultaneously occurs, further improving water collection rate. Those processes require much lower energy consumption than conventional ones. Besides, water collected through guttation do not need further condensation. Therefore, this liquid to liquid steam is much more practical and energy-efficient.

As suggested, we further described the difference and advantages of this device in the discussion part, which is highlighted by yellow.

5) The authors claim a purification rate of $4.2 \text{ kg m}^{-2} \text{ h}^{-1}$. I am not convinced about how this rate is computed. Reading the manuscript, I think that this is the release rate of purified water from the proposed material under sunlight, but the latter has nothing to do with the (continuous) purification rate. In fact, the proposed device cannot continuously operate to generate clean water with a purification rate of $4.2 \text{ kg m}^{-2} \text{ h}^{-1}$. This is clear by looking at Fig. 6, where the waterwheel-like structure is shown: Clearly a big rotating structure is needed for allowing the proposed material to operate. Hence

much larger surface areas are needed for installation and this reduces significantly the continuous purification rate per unit of surface area. Please correct this point in your abstract for avoiding misleading comparisons with other devices, which are designed for real continuous operation.

Reply: It is a really good point. Truly, as you suggested, the purification rate should be better called release rate or collection rate. This is calculated from a single PNPG-F purifier. The device shown in Fig. 6 is an artificial equipment which is used for continues clean water production. For avoiding misleading, we have substituted collection rate for the purification rate both in the revised manuscript.

We discussed the details of computation of the collection rate in the revised manuscript (Supporting information, pages 4-5, and Supplementary Figure S17). For your convenience, we list the details as following for your review. According to your advices, further modification of the name is also made. The release rate of purified water from the proposed material under sunlight is consisting of two part *e.g.* water collected by transpiration and guttation. The experiments were conducted at an ambient temperature of ~ 23 °C and a humidity of $\sim 65\%$. The water collection experiments were performed under the illumination of a solar simulator (CEL-HXF300). As soon as turning up the light, the mass change was immediately tracked real-time by an electronic mass balance communicated to a computer. In case of guttation efficiency, the oozed water was drained through a channel (Figure Ra). The whole weight loss over the entire process arose from both transpiration and guttation. As shown in Figure Rb, in refer to the transpiration efficiency, the samples were floated in a beaker. The oozed water was recollected by the water container. The weight loss arose only from solar heating evaporation. The evaporation rates in the dark field were subtracted from all of the measured evaporation samples. In order to avoid the influence of shape change and surface determination on the calculation result, we prepared series of sound shaped samples with diameter of 90 mm (the diameter of sunlight spot is 60 mm, Figure Rc). The effective illumination area only depends on the sunlight spot, avoiding the effect of size change of samples on the calculation. Only the upper surface samples were coated

with PG to avoid the heating of the side surface (Figure Rd). We have added the above details in the revised manuscript.

Figure R. Schematic illustration of the method to calculate the water generation rate. a, transpiration and guttation; b transpiration; c and d, Digital images of PNPg-F, c (top-view), d (side-view).

6) The authors claim that “As long as the water collection behaviour is the condensation of steam, the efficiency is pinned near the efficiency limits (ca. $1.5 \text{ kg m}^{-2} \text{ h}^{-1}$)”. This is not true. The condensation energy can be fruitfully re-used multiple times, as far as the corresponding temperature is above the ambient one. See the paper by Eliodoro Chiavazzo et al., *Nature Sustainability*, vol 1, pages 763–772 (2018), <https://doi.org/10.1038/s41893-018-0186-x>.

Reply: We fully agree with this point. The corresponding reference have been added in the revised manuscript. And we further corrected the related description in the revised version.

7) There are some relevant papers missing in the list of references, e.g. the pioneering paper by Hadi Ghasemi et al., *Nature Communications*, vol 5, 4449 (2014), <https://doi.org/10.1038/ncomms5449>. Maybe the authors could refer to some recent review papers, e.g. Chen et al., “Challenges and Opportunities for Solar Evaporation”, *Joule* (2018), <https://doi.org/10.1016/j.joule.2018.12.023> or Wang, P., “Emerging investigator series: the rise of nano-enabled photothermal materials for water evaporation and clean water production by sunlight,” *Environmental Science: Nano*,

Reply: Thanks greatly for providing these valuable reports. We have added them in the revised manuscript.

Minor:

The authors say that “According to Fig. 3d, the ratio of transpiration and guttation in case of PNPG-F purifier are 75% and 25%, respectively”. Isn't the other way around? I mean guttation is 75%, right?

Reply: Thanks a lot for reminding of the referee. It is true that the ratio of guttation under this condition is 75%. We have corrected it in the revised manuscript.

List of changes:

1. Page 1, “high rate ($4.2 \text{ kg m}^{-2} \text{ h}^{-1}$) of clean water production” was changed to “high water collection rate ($4.2 \text{ kg m}^{-2} \text{ h}^{-1}$) for a single PNPG-F”.
2. Pages 5 and 6, a paragraph was added to discuss the osmotic pressure of the hydrogel.
3. Page 8, “phase change” was changed to “drastic water content change”.
4. Page 9, “75% and 25%” was changed to “25% and 75%”.
5. Page 10, Three sentences were added to classify the mechanism for guttation.
6. Page 6, three sentences were added to classify how the device avoid membrane contamination.
7. Page 16, we added more description to show difference of this design from conventional solar steam generator.

Additional references:

1. Ref. 5-7, three more references were cited to give a more comprehensive understanding of the corresponding field.
2. Ref. 9, a recent paper was cited to show the challenges of solar steam generation and condensation of steam.
3. Ref. 25-27, previous reports showing the osmotic pressure of the hydrogel in case of forward osmosis.

REVIEWERS' COMMENTS:

Reviewer #4 (Remarks to the Author):

The authors replied satisfactorily to my previous comments. Hence the manuscript seems now ready for publication.

Minor comment: At least citations 5 and 9 show some mistakes. The authors switched names and surnames of cited authors. For example, the first name is Hadi and the surname is Ghasemi. Hence the Ref. 5 should be H. Ghasemi et al., "Solar steam...". Similarly Ref. 9 should be E. Chiavazzo et al.. Please check all your references with this regards.

Reply: Thank you, we edited the format of those two citations and rechecked all the other citations in the revised manuscript.